# Bioaccessibility and Intestinal Transport of Tebuconazole in Table Grape by Using In Vitro Digestion Models

**DOI:** 10.3390/foods11233926

**Published:** 2022-12-05

**Authors:** Xiaowei Liu, Ying Han, Ouli Xiao, Weiye Cui, Jieyin Chen, Xiaofeng Dai, Minmin Li, Zhiqiang Kong

**Affiliations:** 1State Key Laboratory for Biology of Plant Diseases and Insect Pests, Institute of Plant Protection, Chinese Academy of Agricultural Sciences, Beijing 100193, China; 2Western Agricultural Research Center, Chinese Academy of Agricultural Sciences, Changji 831100, China; 3Key Laboratory of Agro-Products Quality and Safety Control in Storage and Transport Process, Ministry of Agriculture and Rural Affairs/Institute of Food Science and Technology, Chinese Academy of Agricultural Sciences, Beijing 100193, China

**Keywords:** grape, fungicide, in vitro model, bioaccessibility, Caco-2 cells

## Abstract

In this study, the effects of various digestive models, influencing factors and dietary supplements on the bioaccessibility of tebuconazole in table grapes were compared. The Caco-2 cell model was employed to reveal the transfer behavior of tebuconazole. The results indicated that digestion time is the main factor affecting bioaccessibility. With an increase in time, the tebuconazole in grapes was almost completely dissolved, with bioaccessibility reaching 98.5%, whereas dietary fiber reduced bioaccessibility. Tebuconazole undergoes carrier-free passive transport in permeable cells in the Caco-2 cell model. These findings have practical application value for correctly evaluating the harmful level of pollutants in the matrix to human body.

## 1. Introduction

Grapes (*Vitis vinifera*) are a nutritious and favored fruit that is rich in vitamins, minerals, oxalic acid, dietary fiber, fat, and polyphenols [1]. Not only can grapes be eaten raw, but they can also be processed into various products, such as juice, raisins, wine, grape seed oil, and jams [2]. However, during planting, grapes are susceptible to various fungal diseases, including powdery and downy mildew, anthracnose, and black rot [3], seriously impacting grape growth, yield, and characteristics [4]. Therefore, fungicides have become the easiest and most effective way to prevent and treat these diseases. Tebuconazole is a triazole fungicide commonly used on grapes, and it has the advantages of a wide sterilization range, rapid action, and low drug resistance [5]. However, even if tebuconazole is applied in accordance with Good Agricultural Practices, its residues may be carried into the table grapes [6].

Bioaccessibility refers to the proportion of contaminants released from the matrix into the digestive solution of the gastrointestinal tract [7]. Assessing the dietary exposure risk of foods by combining in vitro exposure with in vivo digestion can also more truly assess the amount of pesticides absorbed by the human body [8]. However, animal experiments have shown that their individual differences are costly, time intensive, and often curtailed for humanitarian reasons [9]. In contrast, in vitro simulated digestion has high repeatability, a short cycle, and mimics the human body. Therefore, it is a more suitable method for assessing pollutants, heavy metals, and other substances. In addition, the amount of digestive juice used in this method is similar to that which is absorbed in the human body; hence, this method does not over- or underestimate harm to the body [10].

For in vitro simulated digestion, the commonly used static methods include the Deutsches Institut für Normung e.V. (DIN) method [11], in vitro gastrointestinal extraction method (IVG) [12], physiologically based extraction test (PBET), colon-extended PBET (CE-PBET) [13], Solubility Bioaccessibility Research Consortium (SBRC) assay [14], simulator of human intestinal microbial ecosystem (SHIME) [15], the Rijksinstituut voor Volksgezondheid en Milieu (RIVM) method [16], unified bioaccessibility research group of Europe method (UBM) [17], fed organic estimation human simulation test (FOREhST) [18], and INFOGEST (an standard static in vitro digestion model) [19]. The DIN, original SBRC, and IVG methods are similar in their application. DIN is a standardized method for determining organic and inorganic substances in the soil. After SBRC is standardized, it is often used to determine the bioaccessibility of heavy metals. IVG and PBET involve a digestion simulation for environmental pollutants. CE-PBET is similar to PBET in that the digestion time in the intestine is longer. SHIME not only simulates inorganic salts, but also incorporates various digestive enzymes. The RIVM method has also been established for the in vitro digestion simulation of heavy metals but is less used due to its complexity and inaccuracy. Recently, the FOREhST method has been specifically developed for the simulated digestion of food. UBM includes the process of oral digestion, but the bioaccessibility obtained is low. Therefore, the DIN, IVG, PBET, SBRC, and SHIME methods were selected in this experiment for comparison.

During the simulation of digestion process, the different reagents, pH value, solid–liquid ratio and digestion time of different in vitro simulated digestion methods affect bioaccessibility. Studies have revealed that as pH increases, the change in bioaccessibility becomes uncertain, with the difference being related to the physical and chemical properties of the tested substance [20,21]. When setting the solid–liquid ratio, the nature of the substrate itself should be taken into account, considering that there may be a difference in the intake of soil and the food itself [22,23]. Further, as the digestion time increases, harmful substances are adsorbed or desorbed during the simulated digestion process, showing different experimental results [20]. Therefore, the influence of factor changes during digestion on bioaccessibility is worthy of discussion.

Due to the variety of eating habits, the intake of common diets can also be simulated using ingested foods containing hazardous substances. Studies have demonstrated that dietary fiber can reduce the absorption of heavy metals [24] and promote the release of polyphenols in a simulated digestive environment [25], making our understanding of dietary fibers relative. When protein supplements and foods containing hazardous substances are ingested simultaneously, the results are diverse [15,26]. Adding lipids to food reduces the absorption of harmful lipophilic substances [27]. These complex trends are common in human digestion; hence, it is necessary to comprehensively investigate the changes in bioaccessibility after dietary intake.

The human epithelial colorectal adenocarcinoma cell (Caco-2 cell) model is derived from the human colon and has the same advantages as the small intestine and colon cells in simulating the transport of harmful substances [28]. The transport and absorption test of compound in Caco-2 cell model has a good correlation with human oral absorption test [29] which can be used as a more reliable model of in vivo conditions to evaluate the bioaccessibility at the intestinal level [30]. As Carvret et al. determined, the content of environmental pollutants after cell transport is 2 to 7-fold higher than that of the directly detected value with ultra-performance liquid chromatography (UPLC) [31]. The Caco-2 cell transport model has also demonstrated that mixed pesticides are less harmful to the human body than indicated by the results of direct testing [32]. To obtain results closer to the absorption in the human body, it is necessary to use Caco-2 cells to transport the tested substance and determine the degree of harm and absorption mechanism.

This study explored five common in vitro digestion models, and the method that could maximize simulated bioaccessibility was selected. In addition, the influencing factors, including pH, solid–liquid ratio, and digestion time were explored, and the most influential parameter was established. The aim was to determine the impact of dietary ingredients (dietary fiber, protein, and vegetable oil) on bioaccessibility and reveal the transport mode of tebuconazole in cells. The study demonstrates a realistic human body simulation of the digestion of grapes containing tebuconazole.

## 2. Materials and Methods 

### 2.1. Chemicals and Reagents

Tebuconazole standard solution (purity 99.3%) in acetonitrile with concentration of 1000 µg/mL was obtained from Alta Scientific Co., Ltd. (Tianjin, China). LC-MS grade formic acid and acetonitrile were purchased from Thermo Fisher Scientific (Ottawa, ON, Canada). Cleanert MAS-Q (including 50 mg C18 and 150 mg MgSO_4_) was obtained from Agela Technologies (Tianjin, China), while other chemical reagents (listed in Appendix A) were purchased from Sinopharm Chemical Reagent Co., Ltd. (Shanghai, China) and Sigma-Aldrich (Poole, UK). Ultrapure water was prepared using a Milli-Q system (Millipore Corporation, Bedford, MA, USA).

### 2.2. Sample Preparation

The red globe grape samples collected (containing no tebuconazole as determined by residual analysis) were washed, dried, broken up using a homogenizer (NP-S10; Beijiao, Ningbo, China), and mixed evenly; part of the grape samples was reserved for control and matrix extraction. Other grape samples were spiked with an appropriate amount of tebuconazole standard solution to contain 0.02, 0.2, and 2 mg/kg tebuconazole, respectively. After standing for 1 h, the samples were homogenized at −20 °C and kept for later use. To simulate digestion, the grape sample with the concentration of tebuconazole (2 mg/kg) was prepared. The selection of the target pesticide addition level was based on the Chinese National Food Safety Standard: Maximum residue limits for pesticides in food (GB-2763-2021), where the maximum residual concentration limit of tebuconazole in grapes is 2 mg/kg. Grape samples were purchased from local markets in the Haidian district, Beijing City, China. All samples were homogenized and stored at −20 °C until analysis. 

### 2.3. Tebuconazole Residue Analysis 

#### 2.3.1. Sample Extraction and Purification

Determination of tebuconazole residue was carried out using an improved QuEChERS (Quick, Easy, Cheap, Effective, Rugged, Safe) pretreatment method. Briefly, 10 mL of acetonitrile was added to 10 mL of a sample containing tebuconazole, vortexed, and mixed with 4 g of MgSO_4_ and 1 g of NaCl. After mixing, the sample was centrifuged at 3486× *g* for 5 min, and a 1.5 mL aliquot of the supernatant was transferred to a Cleanert MAS-Q kit (Agela Technologies) and centrifuged at 2400× *g* for 3 min. Thereafter, the supernatant was passed through a 0.22 µm nylon syringe filter and then placed in a brown sample bottle until ultra-high liquid chromatography tandem triple quadrupole mass spectrometry (UPLC-TQ-S) analysis.

#### 2.3.2. Tebuconazole Residue Analysis

An Acquity UPLC^®^ BEH C18 column (1.7 µm; 2.1 × 50 mm; Waters Corporation, Milford, MA, USA) on an Acquity UPLC TQ-S system (Waters Corporation) was used to analyze the concentration of the tebuconazole. The mobile phases A and B were 0.1% formic acid water (*v*/*v*) and acetonitrile, respectively. Gradient elution conditions were as follows: 0 min, 10% A; 1.5 min, 90% A; 3 min, 10% A; 3.1 min, 90% A; total time 5 min. The injection volume was 5 µL, the flow rate was 0.3 mL/min, and the sample injection room temperature was 5 °C. The MRM multi-reaction detection mode and an ionization mode positive ion (ESI+) mode were used; the ion source temperature was set to 150 °C, and the capillary voltage was 3.1 kV. The molecular formula and weight of tebuconazole were C_16_H_22_ClN_3_O and 307.82, respectively. The quantitative and qualitative analysis included *m*/*z* = 124/70, respectively, with a cone voltage of 46 V and collision voltages of 18 and 34 V.

### 2.4. Bioaccessibility Analysis of In Vitro Simulation

The protocols of DIN [11], IVG [12], PBET [13], SBRC [14], and SHIME [15] used in this study were applied with slight modifications, and the detailed parameters were as shown in Appendix A. An appropriate amount of sample containing tebuconazole was placed in a 50 mL centrifuge tube with simulated gastric juice and ventilated nitrogen—to make the environment anaerobic—and placed in a shaker at 100 rpm/min avoiding direct light at a constant temperature of 37 °C. After the simulated gastric digestion was completed, intestinal juice was added, maintained in the same environment, filtered through a 0.45 µm water phase filter membrane, and subjected to QuEChERS treatment for residue detection. The method resulting with the highest yields of bioaccessible tebuconazole was selected for further experiments.

### 2.5. Impact of In Vitro Digestion Model Conditions on Tebuconazole Bioaccessibility

After selecting a suitable in vitro simulated digestion model, the changes in bioaccessibility caused by changes in different influencing factors were explored (pH of digestive juice, solid–liquid ratio, digestion time). The pHs of the gastric phase were 1.3, 1.6, 1.9, 2.2, 2.4, and 2.6, and those of the intestinal phase were 5.9, 6.4, 6.9, 7.4, and 7.9 [33]. For the solid–liquid ratio, the ratio of the matrix to simulated digestive and gastrointestinal juices was controlled at 1, 2, 5, 10, and 20%. The digestion time of the gastric stage was 0.5, 1, 1.5, 2, and 3 h, and that of the intestinal stage was 2, 3, 4, 6, and 8 h. The other conditions of the in vitro simulated digestion process were the same.

### 2.6. Influence of Food Matrix Composition on Tebuconazole Bioaccessibility

We explored the changes in tebuconazole bioaccessibility after spiking grape samples with dietary fiber, protein, and vegetable oil, respectively. The number of dietary ingredients added was based on the scientific dietary intake guidance of the Chinese Residents Dietary Guidelines (2016) and the Chinese Residents Dietary Guidelines Research Report (2021). The amount of dietary fiber powder (Metamucil, Procter & Gamble, Lima, OH, USA) and 100% whey protein powder (Swisse, Melbourne, Australia) added was 0.1, 0.2, 0.5, 1, and 2% of the sample mass ratio, while that of pressing maize oil (Changshou Flower Food Co., Ltd., Qingdao, China) was 0.02, 0.05, 0.2, 0.4, and 1%. The configured samples were processed according to the digestion process, with five replicates in each group.

### 2.7. Transport of Tebuconazole in the Caco-2 Cell Model

#### 2.7.1. Cell Culture

The human colorectal adenocarcinoma cell line Caco-2 (ATCC; HTB-37; DSMZ; ATCC169, ECACC09042001, ECACC; 86010202, Beijing, China) was used in this study. The culture fluid was aspirated from the cell culture flask and allowed to stand in a CO_2_ cell incubator (HERAcell 240i; Thermo Fisher Scientific; Waltham, MA, USA); remaining medium in the bottle was aspirated, and phosphate-buffered saline (PBS; Solarbio Technology Co., Ltd., Beijing, China) was used to wash the cell surface before adding 2 mL 0.25% trypsin-EDTA (Lot no: 2186970; Gibco, Grand Island, NY, USA) to digest the cells for 2–4 min after aspiration. The cell status was observed by inverted microscope (Caikang Optical; Shanghai, China) before adding MEM/EBSS-10 medium (Lot No: AG29584972; Cytiva, Marlborough, MA, USA) and fetal bovine serum (FBS, Lot No: 1907301; *v*/*v*:90/10) to stop the intestinal digestion. Thereafter, the cells were digested in a 100 mm cell culture dish until the cells adhered to the wall and reached a confluence of approximately 80%. This process was three times repeated, and the passaging ratio was 1:3.

#### 2.7.2. Cytotoxicity Assays

Using an MTT kit (Solarbio Technology Co., Ltd., Beijing, China), the appropriate tebuconazole concentration in a Caco-2 model was selected. The cell suspension was adjusted to 1 × 10^5^ cells/mL, and 100 µL was injected per well in a 96-well cell culture plate. After 24 h, the cells had adhered to the wall, and solvents containing tebuconazole of different concentrations were added, as well as two groups of acetonitrile with the same amount of tebuconazole solvent in the largest proportion as a control. After incubating in a cell culture incubator for 24, 48, and 72 h, the absorbance was determined in the FC Multiskan microplate reader (ThermoFisher, Waltham, MA, USA) at 540 nm to calculate cell viability.

#### 2.7.3. Transport Studies

The cell transport test [34] procedure was conducted on the nested polycarbonate membrane of a Transwell (Corning; New York, NY, USA) on an ultra-clean table for 2 h using Rattailtendon collagen type Ⅰ (Solarbio Technology Co., Ltd., Beijing, China) diluted with acetic acid–water according to the manufacturer’s instructions. The remaining collagen was removed and rinsed with PBS thrice to digest the viable cells to 2 mL. Afterward, 0.5 mL of cell suspension was added to the apical (AP) side and 1.5 mL of MEM/EBSS-10 to the basolateral (BL) side. After the third day, the medium was changed every other day and every day after day 8.

A Millicell-S (Cat. No. MERS00002; Millipore Corporation, Bedford, MA, USA) epithelial voltage ohmmeter was used to measure transepithelial resistance (TEER) after each fluid change to monitor the integrity of the Caco-2 cell monolayer. The improved Caco-2 cell transport test method was adopted [34]. After aspirating the cell culture fluid and using PBS buffer to wash the cell layer thrice, an appropriate amount of MEM/EBSS-10 containing tebuconazole was added to the AP side with Hank’s Balanced Salt Solution (HBSS; Solarbio Technology Co., Ltd., Beijing, China) added to the BL side for the transport experiment. The outflow test was opposite the transport experiment. Three parallels in each group were established, and samples were collected at 0.5, 1, 1.5, 2, 3, 4, and 6 h. Ultra-Performance Liquid Chromatography Triple Quadrupole (UPLC TQ-S) was used to determine the specific drug amount.

The physiological characteristics of Caco-2 cells before and after differentiation varied, and insufficiently differentiated cells could not meet the experimental requirements. Therefore, the transmembrane resistance of the cells needed to be tested before the transport test to ensure the integrity and density of the cells, and cell resistance was monitored every day.

### 2.8. Data Analysis

The experiments were conducted in triplicate, and the data are shown as the average value ± standard deviation (SD). We used Waters MassLynx V4.2 (Waters Corporation) to analyze the HPLC-TD-S data, Microsoft Office Excel 2019 for the additive recovery data analysis (Microsoft Corporation; Redmond, WA, USA), and IBM^®^ SPSS^®^ statistics 26.0 (SPSS Inc, Chicago, IL, USA) for one-way ANOVA analysis and Duncan test at the 0.05 significance level. In addition, we mapped the data using Origin Pro 2021 (Origin Lab Corporation, Northampton, MA, USA).

Bioaccessibility at the gastric and intestinal stages was calculated using the following equation: Bioaccessibility (BA, %) = *C*_1_*V*/*CM* × 100%,(1)
where *C*_1_ is the concentration of tebuconazole in digestive juice (mg/kg), *V* is the volume of digestive juice (mL), *C* is the added concentration of tebuconazole in grapes (mg/kg), and *M* is the quantity of grapes added to the digestive system (g). The apparent permeability coefficient of tebuconazole in Caco-2 cell model transport was calculated as follows: Permeability coefficients (P_app_) = *(dQ/dt*)/*AC*_0_ × 100%,(2)
where *dQ* is the amount of tebuconazole on the receiving side, *dt* is the transit time, *A* is the surface area of cells on the Transwell membrane, and *C*_0_ is the concentration of tebuconazole.
Permeability directivity (PDR) = Papp (AP-BL)/Papp (BL-AP),(3)
where the Papp (AP-BL) is for the AP side switch to the side of the BL apparent permeability coefficient (cm/s), Papp (BL-AP) is the apparent permeability coefficient (cm/s) of BL side to AP side.

## 3. Results and Discussion 

### 3.1. Method Validation

Our study findings were evaluated based on selectivity, linearity, accuracy, precision, the limit of detection (LOD), and the limit of quantification (LOQ). First, standard solution of tebuconazole was used for establishing standard curve. The *R*^2^ > 0.9997 indicates that the tebuconazole residue analysis in grape samples had good linearity. The test results of additive recovery (Appendix A) were at the additive level of 0.02, 0.2, and 2 mg/kg, the average recovery was 71.10–116.80%, the relative standard deviation (RSD) was ≤11.38%, and *R*^2^ was ≥0.9910. The LOD and LOQ estimated using signal–noise ratios (S/N) of 3 and 10 were less than or equal to 0.02 and 0.05 µg/kg, respectively. These results indicate that the accuracy and precision levels of pesticide analysis were satisfactory.

### 3.2. Bioaccessibility of Various In Vitro Simulated Digestion Methods

The different in vitro simulated digestion methods that have been established to test the bioaccessibility of different substances have different parameters, including composition, pH value, solid-liquid ratio, and time. Here, we partially modified the original parameters in five common digestion models and obtained the bioaccessibility of the gastric and intestinal stages of the five in vitro simulated digestion methods (Table 1). For all the methods, the bioaccessibility values of the intestinal stage were higher than those of the gastric stage. Among them, the bioaccessibility calculated using the SBRC method was the highest in the gastric and intestinal stages (=50.02 ± 8.95 and 76.73 ± 2.80, respectively). Therefore, these results indicated that the SBRC method has a good effect on the extraction of tebuconazole from grapes, it is a relatively conservative assessment of the edible safety of pesticide residues in grapes and was thus used as the basic simulated digestion model for the subsequent variable exploration.

### 3.3. Factors Affecting Bioaccessibility Based on the SBRC Method

The influence of pH, solid–liquid ratio, and digestion time in the gastric and intestinal stages was evaluated, and their impact on bioaccessibility was assessed using the SBRC model. As shown in Figure 1a, with an increase in pH, the gastric stage first declines and then stabilizes, with the highest bioaccessibility (69.99%) at pH 1.3. The intestinal stage first stabilized, decreased, and then remained steady, with the highest bioaccessibility of 47.70% at pH 5.9. This trend may be due to pancreatin having a stronger binding force or degradability in the simulated intestinal juice, which lowers the bioaccessibility value of digested matter after entering the intestinal stage [22]. In addition, pH changes the physical and chemical properties of harmful substances, affecting bioaccessibility [35]. To maximize simulated human body absorption, pH values of 1.3 and 5.9 in the intestinal stomach stages were selected as the pHs of the simulated liquid in subsequent experiments.

The solid–liquid ratio refers to the ratio of the ingested substance to the digestive juice (Figure 1b). The bioaccessibility of the gastric and intestinal stages from 1:100 to 1:5 in the solid–liquid ratio increases first and then decreases. The bioaccessibility is highest at 1:20 (gastric phase: 54.28%; intestinal phase: 81.84%). As the solid–liquid ratio increases, the tebuconazole in grapes can be better dissolved in the digestive juice, and the bioaccessibility also increases. With the increase in grape mass (solid–liquid ratio > 20:1), the contact surface between grape and digestive juice decreases, and the dissolution of tebuconazole decreases, thus the bioavailability decreases. Finally, the gastric and intestinal phases of 1:20 were selected as the solid–liquid ratio for the follow-up test.

With increasing digestion time (Figure 1c), bioaccessibility gradually increased in both phases. The gastric phase increased with time; however, the difference in bioaccessibility between the groups was insignificant. The bioaccessibility of the intestinal phase did not significantly differ between 3, 4, and 6 h. When digestion time reached 8 h, the tebuconazole in the grapes was almost completely dissolved (98.5%). These findings are similar to previous research [23] relating to the balance between adsorption and desorption [12]. Finally, in the subsequent experiments, the digestion times in the gastric and intestinal phases were 1 and 3 h, respectively. 

According to the above experimental results, simulated intestinal digestion at pH 1.3 in the gastric phase, 1 h digestion, pH 5.9 in the intestinal phase, and a solid–liquid ratio of 1:20 were the basic conditions for the following dietary supplement test.

### 3.4. Changes in Bioaccessibility Caused by the Addition of Dietary Ingredients

When the human body ingests food, the different consistency of dietary components in various foods impact bioaccessibility of pesticides in different ways [26].

The addition of dietary fiber to the matrix reduces the overall bioaccessibility of tebuconazole in grapes during the stomach and intestinal phases. Figure 2a shows that the highest value of bioaccessibility of tebuconazole (28.52%) in the gastric phase occurs when the mass ratio of dietary fiber to tebuconazole-containing grapes is 0.1%. When the ratio increases to 1:50, the bioaccessibility decreases to 13.58%. Compared with the average bioaccessibility in the gastric phase (24.31%), the average bioaccessibility in the intestinal phase (49.95%) was twice that of the gastric phase. This is primarily attributed to the acidic conditions. The soluble pectin in dietary fiber encapsulates the matrix, resulting in lower bioaccessibility. In the intestinal phase, adding bile salt and pancreatin produces a surfactant effect to dissolve the tebuconazole in food, then improving bioaccessibility [18].

Overall bioaccessibility following simulated digestion in the gastric phase was higher than that in the intestinal phase. Figure 2b shows that the amount of protein added during simulated digestion has a small effect on bioaccessibility, but the overall increase in the treatment demonstrates an upward trend consistent with previous studies [36]. The bioaccessibility in the gastric phase was 82.17–87.10%, and that in the intestinal phase was 65.05–73.10%. This may be related to the large amount of tartaric acid, fruit acid, and other acidic substances in grapes, the low pH of the simulated gastric juice, and protein denaturation in an over-acidic environment [37]. Therefore, a large amount of tebuconazole dissolves in the gastric phase, and bioaccessibility is high. In the intestinal digestion phase, pancreatin helps absorb and transform denatured protein, reducing the bioaccessibility of tebuconazole.

The bioaccessibility of tebuconazole after the vegetable oil addition displayed remarkable difference (Figure 2c). In the gastric phase, with increasing vegetable oil addition, bioaccessibility decreased from 57.78 to 46.67%. In the intestinal phase, the added amounts were 0.02, 0.05, 0.2, 0.4, and 1%, showed that the presence of vegetable oil has impact on the bioaccessibility of tebuconazole. Compared with the original method, adding vegetable oil reduced tebuconazole bioaccessibility, similar to previous studies [38]. This may be indicative of the fact that the oil-based surface activation of vegetable oils can effectively prevent pesticide migration.

Adding dietary fiber can significantly reduce the bioaccessibility of tebuconazole. The effect of vegetable oil is weak, and that of protein is complicated. Therefore, increasing dietary fiber intake plays an important role in reducing pesticide absorption.

### 3.5. Caco-2 Cell Model Analysis of Intestinal Absorption of Tebuconazole

To further investigate the process of tebuconazole absorption and transport in the intestinal tract, a Caco-2 cell model was used. After 18 days of cell differentiation, the resistance value was (TEER) > 500 Ω/cm^2^ and remained unchanged between 18 and 21 d, meeting the requirements of the test [34].

Exogenous substances affect cell viability at a certain concentration. When testing for the effects of tebuconazole on cell viability, the same amount of solvent was used at the maximum dose. As shown in Figure 3, different concentrations of acetonitrile and tebuconazole used in the experiment did not affect Caco-2 cell viability. The concentration of tebuconazole in the transfer test was higher than that of other triazole substances in the transfer test [33], which is likely due to the nature of tebuconazole [39]. The tendency of pesticide bioaccessibility may depend on the polarity of pesticide molecules, and the bioaccessibility increases with the increase in pesticide polarity.

The apparent permeability coefficient (P_app_) and permeability directivity (PDR) of tebuconazole in Caco-2 cells were also analyzed (Table 2). The permeability demonstrated a positive correlation with changes in time. The average apparent permeability coefficient (P_app_) at the AP-BL terminal was 2.87 × 10^−5^ cm/s, and that of the BL-AP terminal was 2.86 × 10^−5^ cm/s. The apparent permeability coefficient (P_app_) was >1 × 10^−5^ cm/s, indicating that tebuconazole had a good absorption capacity in Caco-2 cells [34]. The PDR was 1.01, and the transfer and outflow coefficients at AP and BL ends were similar. This PDR value (0.5–1.5) indicated that tebuconazole is a pesticide with similar transport and outflow capacity with no carrier involved in its transportation, suggesting a simple transcellular passive method [34]. 

## 4. Conclusions

This study compared tebuconazole bioaccessibility in several in vitro simulation methods and explicit effects of different dietary components on the bioaccessibility of tebuconazole in grape. The SBRC method exhibited the highest bioaccessibility in the gastric and intestinal stages. When the pH, solid–liquid ratio, and time change, bioaccessibility changes; for example, bioaccessibility gradually decreases with increasing pH. When time is used as a variable, the opposite is true. A solid–liquid ratio of 1:20 could provide higher bioaccessibility. The intake of dietary fiber could significantly reduce the absorption of tebuconazole; it is suggested that people should moderately strengthen the intake of dietary nutrients in their daily life. Furthermore, the Caco-2 cell model verified that tebuconazole is passively transported in the intestine. Our findings suggest that the bioaccessibility of pesticide residues needs to be taken into account when assessing the risk of dietary exposure to pesticides in the future.

## Figures and Tables

**Figure 1 foods-11-03926-f001:**
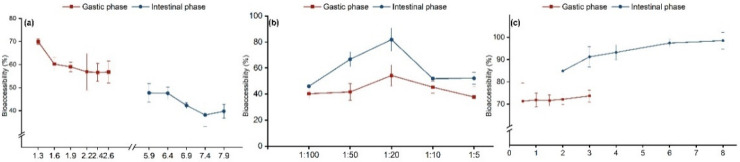
Tebuconazole concentration changes in grapes at different pH values (**a**), solid–liquid (S/L) ratios (**b**), and digestion times (**c**). Error bars represent the standard deviation of five replicates.

**Figure 2 foods-11-03926-f002:**
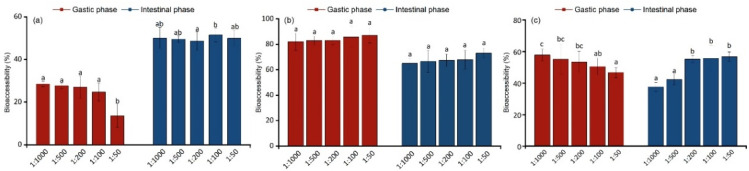
The effect of adding dietary fiber (**a**), protein (**b**), and vegetable oil (**c**) to grapes containing tebuconazole on bioavailability. Error bars represent the standard deviation of five replicates. Different lowercase letters at the top of the column indicate significant differences in bioaccessibility, with a *p*-value of 0.05.

**Figure 3 foods-11-03926-f003:**
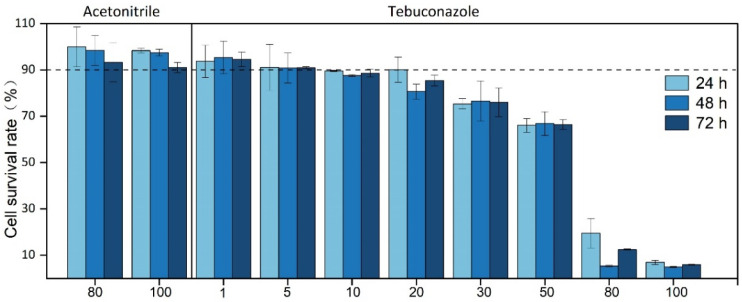
The effect of different concentrations of acetonitrile and tebuconazole on the cell survival rate in the Caco-2 cell model at 24, 48, and 72 h. Error bars represent the standard deviation of five replicates.

**Table 1 foods-11-03926-t001:** Comparison of bioaccessibility of various digestion methods.

Simulation Methods	Bioaccessibility (%)	
	Gastric Phase (%)	Intestinal Phase (%)
DIN	27.47 ± 1.53	72.90 ± 7.77
IVG	33.32 ± 3.56	69.48 ± 3.89
PBET	33.53 ± 5.98	70.00 ± 2.61
SBRC	50.02 ± 8.95	76.73 ± 2.80
SHIME	31.13 ± 1.78	54.83 ± 5.58

**Table 2 foods-11-03926-t002:** Changes of Tebuconazole Transport in Caco-2 Cell Model (n = 3).

Time (h)	Apparent Permeability Coefficient (P_app_)	
	P_app_ AP-BL (1 × 10^−5^ cm^2^/s)	P_app_ BL-AP (1 × 10^−5^ cm^2^/s)
0.5	1.36 ± 0.14	1.71 ± 0.03
1	1.61 ± 0.13	1.79 ± 0.13
1.5	2.13 ± 0.07	2.00 ± 0.40
2	2.15 ± 0.05	2.62 ± 0.09
3	2.66 ± 1.08	3.07 ± 0.44
4	3.84 ± 0.55	3.09 ± 0.12
6	6.37 ± 0.65	5.72 ± 1.82

## Data Availability

Data is contained within the article or Appendix A.

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
