# Peer review of "Bioaccessibility and Intestinal Transport of Tebuconazole in Table Grape by Using In Vitro Digestion Models"

_foods, 2022, doi:10.3390/foods11233926_

Round 1
Reviewer 1 Report
please see attached pdf

Author Response
- Line 35, small "s"
Answer: The uppercase S has been changed to the lowercase S.
- Line 37, syntax/grammar
Answer: Thanks for your comments, we have removed the sentence according to the comments.
- Line 68, one protocol missing, the INFOGEST, which elaborated an international consensus static digestion method to harmonize experimental conditions and has general guidelines, thus allowing the comparison of studies and data.
Furthermore, it is not clear the justification to use all these different models for comparison, especially the one that do not contain gastric enzymes or intestinal enzymes, in other words protocols that do not mimic human conditions. Please explain why not use on standard protocol and test it for different durations, but instead compare all these different protocols.
Answer: Thanks for your comments. By using different model parameters, the static model presents different degrees of difference. INFOGEST in vitro simulation digestion scheme based on the consensus of international experts in multiple fields adopts standardized static digestion method, which is simple to use, but not suitable for the simulation of digestive dynamics, and the obtained biological availability is low. So we not use on standard protocol and test it for different durations, but instead compare all these different protocols.
- Line 102, but is this method close to the true value of bioaccessibility?
Answer: Caco-2 cell model is widely used to study the transport and absorption of foreign substances because it is similar to human intestinal epithelial cells in morphology and biological properties. Under different transport conditions, the experimental data obtained from Caco-2 cell model had a good correlation with the in vivo experimental results.
Borchardt, R. T. (2011). Hidalgo, I. J., Raub, T. J., and Borchardt, R. T.: Characterization of the Human Colon Carcinoma Cell Line (Caco-2) as a Model System for Intestinal Epithelial Permeability, Gastroenterology, 96, 736–749, 1989—The Backstory. The AAPS Journal, 13(3), 323–327. doi:10.1208/s12248-011-9283-8.
- Line 169, please provide reference for pH of intestine 7.9.
Answer: Shi, Y.-H., Xiao, J.-J., Feng, R.-P., Liu, Y.-Y., Liao, M., Wu, X.-W., … Cao, H.-Q. (2017). Factors Affecting the Bioaccessibility and Intestinal Transport of Difenoconazole, Hexaconazole, and Spirodiclofen in Human Caco-2 Cells Following in Vitro Digestion. Journal of Agricultural and Food Chemistry, 65(41), 9139–9146. doi:10.1021/acs.jafc.7b02781.
- Line 242, maybe "quantity"?
Answer: We have now revised the “quality” to “quantity” in line 243.
- Line 270, why this method maximized the results. Is it expected to do so? Is it physiologically relevant? Please explain/elaborate
Answer: Thanks for your comments, we have revised the sentence as:” Therefore, these results indicated that the SBRC method has a good effect on the ex-traction of tebuconazole from grapes, it is a relatively conservative assessment of the edible safety of pesticide residues in grapes and was thus used as the basic simulated digestion model for the subsequent variable exploration” in the manuscript.
- Line 293, why when the concentration of tebuconazole is low in digestive juice there is poor solubility, and when it is higher the solubility improves? Usually, it is more difficult to dissolve a substance when its concentration increases.
Answer: thanks for your comments, with the increase of grape mass (solid-liquid ratio >20:1), the contact surface between grape and digestive juice decreases, and the dissolution of tebuconazole decreases, thus the bioavailability decreases. We have re-organized the sentence.
- Line 373, please elaborate
Answer: Thanks for your comments, we have elaborated the relevant content.

Reviewer 2 Report
1. TITLE
· The title should be rephrased because in this form it is confusing (indicating that there is such a thing as "table grape in vitro model")
· Authors who contributed equally to the manuscript shouldnt be marked with the superscript 4; Their names should be stated in the sentence: „ Xiaowei Liu and Ying Han contributed equally to the preparation of the manuscript“ without the superscript.
2. ABSTRACT
Line 15: The sentence „In this study, the bioaccessibility of tebuconazole in table grapes was investigated utilizing various digestion models, influencing factors, and dietary supplements“ should be modified. One doesnt use influencing factor or dietary supplements to investigate bioacessibility. Please rephrase to achieve adequate accuracy and clarity.
Line 21: The sentence „These findings provide a more realistic pesticide residue risk assessment“ should be rephrased. This study did not provide any data regardnig exposure risk assessment. Please rephrase to achieve adequate accuracy and clarity.
3. INTRODUCTION
General remarks. Big parts of Introduction should be rewritten (as stated in the examples below) since numerous sentences are not clear or are confusing. Generally Introduction lacks structure and clarity. The authors should explain the importance of bioaccessibility/bioavailaiblity research in terms of (pesticide) exposure risk assessment. They should explain more clearly why tebuconazole was chosen as the fungicide of interest. English should also be significantly improved.
Line 25: „well-loved“ please rephrase
Line 33. Please rephrase in the way suggested in the comment of the main text
Line 35. The sentence: „Meanwhile, Several studies have demonstrated that tebuconazole can cause serious interference with the growth and development of various organisms [7,8,9]. Hence, the influence of tebuconazole residues on human health is immeasurable“ should be removed from the manuscript. I am not sure if this is relevant. All pesticides have negative effects on living organisms but still they can be used in accordance with good agricultural practice. If residues appear in food, they should be within ranges permitted by law. Also, the second sentence doesnt make any sense. Maybe instead of this sentence you should focuse on possible cases when tebuconazole was found in grapes (or processed products) in concentrations higher than permitted posing additional health risk) Also, please remove the references 7, 8 and 9 if applicable.
Lines 39-49. The whole paragraph should be re-written to achieve adequate clarity and accuracy. I understand what the authors were trying to say, but it should be rewritten. First they should explain the term "bioacessibility", the importance of assesing it, and than the methods and advantages of in vitro approach.
Lines 51-57. Please add INFOGEST method, since it was developed by the consortium of experts taking into account best possible in vivo-in vitro correlations and reproducibility of obtained data. DOI:https://doi.org/10.1038/s41596-018-0119-1
Lines 70-72. The sentence „During the digestion process, not only do these methods result in different bioaccessibilities, but digestion factors such as pH, solid-liquid ratio, and digestion time can also affect bioaccessibility“ should be rephrased. Different digestion factors is how the methods can be differed to eachother. Also, it is not „during digestion process“ but „during the simulation of digestion process“. Please rephrase to achieve adequate accuracy and clarity.
Lines 84-85. The sentence „When protein and foods containing hazardous substances are in-84 gested simultaneously, the results are diverse [18,28]“ should be rephrased. How do you take food and proteins simultaneously? Proteins are in the food. Did You mean protein supplements? It is confusing, please rephrase to achieve adequate accuracy and clarity.
Lines 89-90. The sentence „In the last step of the human digestion process, food mixed with harmful substances reaches the colon“ should be omitted; it is irrelevant in relation to the rest of the text. Caco-2 differentiate int enterocyte-like cells, allowing mimicking of the absorption in the small intestine (not colon!).
Lines 92-95. The sentence „The transportation mode of Caco-2 cells has a significant correlation with the digestion process in the human body and that of some substances [31], which could offer more reliable models of in vivo 94 conditions for the evaluation of bioavailability at the intestinal level [32]“ should be rephrased to achieve adequate accuracy and clarity.
Lines 95-97. The sentence „As Carvret et al. found that the content of environmental pollutants after cell transport is 2–7-fold higher than that obtained with ultra-performance liquid chromatography (UPLC) [33]“ should be rephrased to achieve adequate accuracy and clarity.
4. MATERIALS AND METHODS
General remarks. Considering numerous methodological and analytical steps of the research i strongly suggest that authors prepare graphical flowchart explaining conducted research design (to make it clear that tebuconazole was extracted from untreated samples, digested samples and Caco-2 cells and to clarify how bioaccessibility/permeability values were calculated. To me it is clear cause I conduct similar research but it might be confusing in general.
Lines 118-120. The sentence „Grape samples were purchased from local markets 118 in the Haidian district, Beijing City, China. All samples were homogenized and stored at 119 −20 °C until analysis“ should be moved to the sbeginning of the section 2.2. Sample spiking
Line 121. The title 2.2. Sample spiking should be renamed to 2.2. Sample preparation
Line 122. What is „earth“ grape samples? What does that mean? Please correct.
Line 122 „by residual analysis)..“ should be replaced with „...as determined by residual analysis“
Line 125-127 The sentence „Other grape samples were treated with an appropriate amount of tebuconazole 125 standard solution, and the samples used for method verification were prepared into a 126 grape matrix containing 0.02, 0.2, and 2 mg/kg tebuconazole“ should be replaced by „Other grape samples were spiked with an appropriate amount of tebuconazole standard solution, ato contain 0.02, 0.2, and 2 mg/kg tebuconazole, respectively“
Line 128 What is the meaning of the sentence „To simulate digestion, a tebuconazole sample at a concentration of 2 mg/kg was prepared from the grapes“ I thought you submitted all the grape samples (spiked and unspiked) to the simulation of gastrointestinal digestion (different protocols)? Please clarify.
Line 130 What is „test concentration“?
Lines 158-159 The sentence „Depending on the several existing in vitro simulated digestion methods, such as DIN,IVG, PBET, SBRC, and SHIME, some parameters were adjusted, as shown in Table S1“ should be rephrased. State clearly what methods were used; state that they differ among themselves regarding the composition of digestion fluids, pH, duration of digestion simulation and solid liquid ratio, as presented in Table S1. Also what do you mean parameters were adjusted? Adjusted to what?
Line 165 Add sentence: "The method resulting with the highest yields of bioaccessible tebuconazole was selected for the further experiments"
Line 166 The title „2.5. Influencing factors in the in vitro simulated digestion model“ should be changed to „2.5 Impact of in vitro digestion model conditions on tebuconazole bioaccessibility“ (or something simmilar)
Line 168 „i.e.“ should be removed form the sentence. All the factors that were varied (investigated) should be stated.
Line 172 The sentence „The digestion time separated the gastric stage from the intestinal stage“ should be removed.
Line 176. The title „2.6. Influence of dietary ingredient addition on bioaccessibility“ should be changed to „2.6. Influence of food matrix composition on debuconazole bioaccessibility“
Line 177-185 The section should be changed, to clarify the conducted procedure. Did you spike grape samples with fiber/protein/oil? You should rephrase and make it clear. "Dietary supplementation" can not be used here - it implies giving dietary supplements to humans/animals.
Lines 189-200 Please check carefully this section and correct where needed. There are lot of parts that are not clear. Refere to the comments I gave in the pdf file
Line 204 Remove „transport“
Line 210 Add the name of the town
Line 213 „Transwell“ should be in the brackets, add producer, town, state
Line 216 „good cells“ should be „viable cells“
Line 227 Please also provide full name of the method applied (UPLC-TQ-S)
RESULTS AND DISCUSSION
Line 252-253 The sentence should be changed to: “Standard solution of tebuconazole was used for establishing standard curve“. In general try to avoid constructions „We applied or we developed“ use passive instead, it is more appropriate for scientific text.
Line 276 Remove 3.3.
Lines 289-297 Do you have any explanation on how increased soolvent:solid ratios (>20:1) resulted in lower tebuconzole bioaccessibility in relation to 20:1 ratio?
Lines 314-316 The sentence „When the human body ingests food, it incorporates it with other ingested dietary components, resulting in a synergistic or antagonistic effect on the actual intake of pesti cides [28]“ should be rephrased. Dietary components are food. Different foods are consistent of different dietary components and therefore impact bioacessibility of pesticides in different way.
Lines 318-320 The sentence „Figure 2a shows that the highest value in the gastric phase occurs when the mass ratio of dietary fiber to tebuconazole-containing grapes is 0.1%, and the bioaccessibility is 28.52%.“ should be rephrased. Highest value of what?
Line 320 Remove „addition“
Lines 324-325 The sentence is incorrect. Cellulose is cellulose and pectin is pectin (it can be maybe classified as haemicellulose). There is no pectin in cellulose. Please change it.
Lines 338-339 The sentence: „The bioaccessibility of vegetable oil following dietary ingredient addition is complex (Fig. 2c)“ should be changed. I dont understand wht do you mean. You didnt measure the bioaccessibility of oil? Also, how can bioacessibility be complex? Rephrase.
Lines 340-341 The sentence: „In the intestinal phase, the addition amounts were 0.02, 0.05, 340 0.2, 0.4, and 1%, with significant differences“ should be changed. „addition amounts“ should be changed to „added amounts“. Also, what do you mean by „...with significant differences“? That they influenced bioaccessibility of tebuconazole with significant difference? Rephrase the sentence.
Lines 343-344 Please explain what did you try to say with the sentence: „...tebuconazole has higher lipophilicity and separates water and oil 343 from the digestive juice“? It makes no sence. Rephrase or remove from the text.
Line 344 What lipid membrane? Rephrase.
Lines 348-349 The sentence „Adding dietary fiber can significantly reduce the contribution of tebuconazole to bioaccessibility“ should be rephrased. How can tebuconazole contribute to its own bioaccessibility?
Lines 358-359 The sentence: „To further investigate the process of tebuconazole absorption and transport in the 358 human body, a Caco-2 cell model was used“ must be changed. Caco-2 model can be used to assess intestinal transport/absorption but not the transport in the human body.
Lines 360-365 The paragraph can be removed. This is basic methodological issue and can optionally be moved to Materials and Methods section. Since it is a common knowledge, it is not necessary to keep it at all. In any case it should be removed from results. Only the last sentence, stating the TEER required for conducting the experiments can be left in Results.
Line 366 The sentence „When tebuconazole is tested for cell viability...“ must be changed. One can not test tebuconazole for cell viability!
Lines 366-374 The whole paragraph should be significantly shorter and simplified. In short, you proved that acetonitrille or used tebuconazole concentrations used in the experiment didnt affect Caco-2 cell viability. One sentence like this is enough.
Line 379 What is permeability directivity? You never mentioned it before nor you explained how it was calculated. In order to use it you need to explain the term and methodology it in materials and methods section
Line 382 Explain the meaning of AP-BL
Line 388 Remove „method“. Use „transport mechanism“ instead
Line 388-390 Remove the paragraph.
Line 391. Change the title of Table 2. Doesnt make sense
Please, also check several additional comments in the pdf version of the manuscript that i didnt mention here.

Author Response
- TITLE
- The title should be rephrased because in this form it is confusing (indicating that there is such a thing as "table grape in vitro model")
Answer: Thanks for your comments, we have rephrased the title as “Bioaccessibility and intestinal transport of tebuconazole in table grape by using in vitro digestion models”
- Authors who contributed equally to the manuscript shouldnt be marked with the superscript 4; Their names should be stated in the sentence: „ Xiaowei Liu and Ying Han contributed equally to the preparation of the manuscript“ without the superscript.
Answer: Thanks for your comments, we have stated their names in the sentence.
- ABSTRACT
Line 15: The sentence „In this study, the bioaccessibility of tebuconazole in table grapes was investigated utilizing various digestion models, influencing factors, and dietary supplements“ should be modified. One doesnt use influencing factor or dietary supplements to investigate bioacessibility. Please rephrase to achieve adequate accuracy and clarity.
Answer: Thanks for your comments, we have revised the sentence as “In this study, the effects of various digestive models, influencing factors and dietary supplements on the bioaccessibility of tebuconazole in table grapes were compared.”
Line 21: The sentence „These findings provide a more realistic pesticide residue risk assessment“ should be rephrased. This study did not provide any data regardnig exposure risk assessment. Please rephrase to achieve adequate accuracy and clarity.
Answer: Thanks for your comments, we have rephrased the sentence as “These findings have practical application value for correctly evaluating the harmful level of pollutants in the matrix to human body”.
- INTRODUCTION
General remarks. Big parts of Introduction should be rewritten (as stated in the examples below) since numerous sentences are not clear or are confusing. Generally Introduction lacks structure and clarity. The authors should explain the importance of bioaccessibility/bioavailaiblity research in terms of (pesticide) exposure risk assessment. They should explain more clearly why tebuconazole was chosen as the fungicide of interest. English should also be significantly improved.
Line 25: „well-loved“ please rephrase
Answer: The “well-loved” has been rephrased to “favored”.
Line 33. Please rephrase in the way suggested in the comment of the main text
Answer: Thanks for your comments, we have rephrased the sentence as “even if tebuconazole is applied in accordance....”
Line 35. The sentence: „Meanwhile, Several studies have demonstrated that tebuconazole can cause serious interference with the growth and development of various organisms [7,8,9]. Hence, the influence of tebuconazole residues on human health is immeasurable“ should be removed from the manuscript. I am not sure if this is relevant. All pesticides have negative effects on living organisms but still they can be used in accordance with good agricultural practice. If residues appear in food, they should be within ranges permitted by law. Also, the second sentence doesnt make any sense. Maybe instead of this sentence you should focuse on possible cases when tebuconazole was found in grapes (or processed products) in concentrations higher than permitted posing additional health risk) Also, please remove the references 7, 8 and 9 if applicable.
Answer: Thanks for your comments, we have removed the sentence and the reference 7, 8 and 9.
Lines 39-49. The whole paragraph should be re-written to achieve adequate clarity and accuracy. I understand what the authors were trying to say, but it should be rewritten. First they should explain the term "bioacessibility", the importance of assesing it, and than the methods and advantages of in vitro approach.
Answer: We have re-written the paragraph as follows:
Bioaccessibility refers to the proportion of contaminants released from the matrix into the digestive solution of the gastrointestinal tract [13]. Assessing the dietary exposure risk of foods by combining in vitro exposure with in vivo digestion, can more truly assess the amount of pesticides absorbed by the human body [10]. However, animal experiments have shown individual differences, are costly, time intensive, and often curtailed for humanitarian reasons [11]. In contrast, in vitro simulated digestion has high repeatability, a short cycle, and mimics the human body. Therefore, it is a more suitable method for assessing pollutants, heavy metals, and other substances. In addition, the amount of digestive juice used in this method is similar to that which is absorbed in the human body; hence, this method does not over- or underestimate harm to the body [12].
Lines 51-57. Please add INFOGEST method, since it was developed by the consortium of experts taking into account best possible in vivo-in vitro correlations and reproducibility of obtained data. DOI:https://doi.org/10.1038/s41596-018-0119-1
Answer: Thanks for your comments, we have added the INFOGEST method.
Lines 70-72. The sentence „During the digestion process, not only do these methods result in different bioaccessibilities, but digestion factors such as pH, solid-liquid ratio, and digestion time can also affect bioaccessibility“ should be rephrased. Different digestion factors is how the methods can be differed to eachother. Also, it is not „during digestion process“ but, during the simulation of digestion process“. Please rephrase to achieve adequate accuracy and clarity.
Answer: Thanks for your comments, we have rephrased the sentence as follows:
“During the simulation of digestion process, the different reagents, pH value, solid-liquid ratio and digestion time of different in vitro simulated digestion methods affect bioaccessibility”.
Lines 84-85. The sentence „When protein and foods containing hazardous substances are in-84 gested simultaneously, the results are diverse [18,28]“ should be rephrased. How do you take food and proteins simultaneously? Proteins are in the food. Did You mean protein supplements? It is confusing, please rephrase to achieve adequate accuracy and clarity.
Answer: Thanks for your comments, we have rephrased the sentence as “When protein supplements and foods containing hazardous substances are ingested simultaneously, the results are diverse “.
Lines 89-90. The sentence „In the last step of the human digestion process, food mixed with harmful substances reaches the colon“ should be omitted; it is irrelevant in relation to the rest of the text. Caco-2 differentiate int enterocyte-like cells, allowing mimicking of the absorption in the small intestine (not colon!).
Answer: Thanks for your comments, we have omitted the sentence in the manuscript.
Lines 92-95. The sentence „The transportation mode of Caco-2 cells has a significant correlation with the digestion process in the human body and that of some substances [31], which could offer more reliable models of in vivo 94 conditions for the evaluation of bioavailability at the intestinal level [32]“ should be rephrased to achieve adequate accuracy and clarity.
Answer: Thanks for your comments, we have rephrased the sentence as “The transport and absorption test of compound in Caco-2 cell model has a good correlation with human oral absorption test [31]. which can be used as a more reliable model of in vivo conditions to evaluate the bioaccessibility at the intestinal level [32]”
Lines 95-97. The sentence „As Carvret et al. found that the content of environmental pollutants after cell transport is 2–7-fold higher than that obtained with ultra-performance liquid chromatography (UPLC) [33]“ should be rephrased to achieve adequate accuracy and clarity.
Answer: Thanks for your suggestion, we have revised the sentence to “As Carvret et al. found that the content of environmental pollutants after cell transport is 2-7-fold higher than that directly detected value with ultra-performance liquid chromatography (UPLC) [33]”.
- MATERIALS AND METHODS
General remarks. Considering numerous methodological and analytical steps of the research i strongly suggest that authors prepare graphical flowchart explaining conducted research design (to make it clear that tebuconazole was extracted from untreated samples, digested samples and Caco-2 cells and to clarify how bioaccessibility/permeability values were calculated. To me it is clear cause I conduct similar research but it might be confusing in general.
Lines 118-120. The sentence „Grape samples were purchased from local markets 118 in the Haidian district, Beijing City, China. All samples were homogenized and stored at 119 −20 °C until analysis“ should be moved to the sbeginning of the section 2.2. Sample spiking
Answer: Thanks for your comments, we have removed the sentence to the section 2.2.
Line 121. The title 2.2. Sample spiking should be renamed to 2.2. Sample preparation
Answer: Thanks for your comments, we have renamed the title 2.2. Sample spiking to 2.2. Sample preparation.
Line 122. What is „earth“ grape samples? What does that mean? Please correct.
Answer: Thanks for your kind observation, we have revised “red earth grape” to “red globe grape” in the manuscript.
Line 122 „by residual analysis)..“ should be replaced with „...as determined by residual analysis“
Answer: Thanks for your suggestion, we have revised the sentence.
Line 125-127 The sentence „Other grape samples were treated with an appropriate amount of tebuconazole 125 standard solution, and the samples used for method verification were prepared into a 126 grape matrix containing 0.02, 0.2, and 2 mg/kg tebuconazole“ should be replaced by „Other grape samples were spiked with an appropriate amount of tebuconazole standard solution, ato contain 0.02, 0.2, and 2 mg/kg tebuconazole, respectively“
Answer: Thanks for your comments, we have replaced the sentence according to your suggestion by “Other grape samples were spiked with an appropriate amount of tebuconazole standard solution, to contain 0.02, 0.2, and 2 mg/kg tebuconazole, respectively”.
Line 128 What is the meaning of the sentence „To simulate digestion, a tebuconazole sample at a concentration of 2 mg/kg was prepared from the grapes“ I thought you submitted all the grape samples (spiked and unspiked) to the simulation of gastrointestinal digestion (different protocols)? Please clarify.
Answer: We have now revised the sentence in the manuscript as follows: “To simulate digestion, the grape sample with the concentration of tebuconazole (2 mg/kg) was prepared”.
Line 130 What is „test concentration“?
Answer: We have revised the “test concentration” to “target pesticide addition level” in the manuscprit
Lines 158-159 The sentence „Depending on the several existing in vitro simulated digestion methods, such as DIN,IVG, PBET, SBRC, and SHIME, some parameters were adjusted, as shown in Table S1“ should be rephrased. State clearly what methods were used; state that they differ among themselves regarding the composition of digestion fluids, pH, duration of digestion simulation and solid liquid ratio, as presented in Table S1. Also what do you mean parameters were adjusted? Adjusted to what?
Answer: thanks for your comments, we have modified the sentence and the detailed parameters of the protocols used in this study were provided in the table S1.
Line 165 Add sentence: "The method resulting with the highest yields of bioaccessible tebuconazole was selected for the further experiments"
Answer: Thanks for your suggestion, we have added the sentence in the manuscript.
Line 166 The title „2.5. Influencing factors in the in vitro simulated digestion model “should be changed to „2.5 Impact of in vitro digestion model conditions on tebuconazole bioaccessibility“ (or something simmilar)
Answer: We have changed the title to “2.5. Impact of in vitro digestion model conditions on tebuconazole bioaccessibility”.
Line 168 „i.e.“ should be removed form the sentence. All the factors that were varied (investigated) should be stated.
Answer: Thanks for your suggestion, we have removed the “i.e.”
Line 172 The sentence „The digestion time separated the gastric stage from the intestinal stage“ should be removed.
Answer: Thanks for your suggestion, we have removed the sentence.
Line 176. The title „2.6. Influence of dietary ingredient addition on bioaccessibility“ should be changed to „2.6. Influence of food matrix composition on debuconazole bioaccessibility“
Answer: We have changed the title to “2.6. Influence of food matrix composition on tebuconazole bioaccessibility”.
Line 177-185 The section should be changed, to clarify the conducted procedure. Did you spike grape samples with fiber/protein/oil? You should rephrase and make it clear. "Dietary supplementation" can not be used here - it implies giving dietary supplements to humans/animals.
Answer: Thanks for your comments, we have rephrased the sentence as:” We explored the changes in tebuconazole bioaccessibility after spiking grape samples with dietary fiber, protein, and vegetable oil, respectively”.
Lines 189-200 Please check carefully this section and correct where needed. There are lot of parts that are not clear. Refere to the comments I gave in the pdf file
Answer: Thanks for your comments, we have checked this section and correct it according to your suggestions.
Line 204 Remove „transport“
Answer: We have removed the word “transport”
Line 210 Add the name of the town
Answer: We have add the town “Waltham”.
Line 213 „Transwell“ should be in the brackets, add producer, town, state
Answer: Thanks for your comments, we have added the corresponding information in the manuscript.
Line 216 „good cells“ should be „viable cells“
Answer: We have revised the “good cells” to “viable cells”.
Line 227 Please also provide full name of the method applied (UPLC-TQ-S)
Answer: We have added the full name of the method applied (UPLC-TQ-S).
5.RESULTS AND DISCUSSION
Line 252-253 The sentence should be changed to: “Standard solution of tebuconazole was used for establishing standard curve“. In general try to avoid constructions „We applied or we developed“ use passive instead, it is more appropriate for scientific text.
Answer: Thanks for your comments, we have revised the sentence according to your suggestion as:” Standard solution of tebuconazole was used for establishing standard curve”.
Line 276 Remove 3.3.
Answer: We have removed 3.3.
Lines 289-297 Do you have any explanation on how increased soolvent:solid ratios (>20:1) resulted in lower tebuconzole bioaccessibility in relation to 20:1 ratio?
Answer: “With the increase of grape mass (solid-liquid ratio >20:1), the contact surface between grape and digestive juice decreases, and the dissolution of tezolol decreases, thus the bioavailability decreases”. We have now added the explanation in the manuscript.
Lines 314-316 The sentence „When the human body ingests food, it incorporates it with other ingested dietary components, resulting in a synergistic or antagonistic effect on the actual intake of pesti cides [28]“ should be rephrased. Dietary components are food. Different foods are consistent of different dietary components and therefore impact bioacessibility of pesticides in different way.
Answer: Thanks for your comments, we have rephrased the sentence as “When the human body ingests food, the different consistent of dietary components in various foods impact bioaccessibility of pesticidests in different way [28] “in the manuscript.
Lines 318-320 The sentence „Figure 2a shows that the highest value in the gastric phase occurs when the mass ratio of dietary fiber to tebuconazole-containing grapes is 0.1%, and the bioaccessibility is 28.52%.“ should be rephrased. Highest value of what?
Answer: Thanks for your commets, we have rephrased the sentence as the follows:
Figure 2a shows that the highest value of bioaccessibility of tebuconazole (28.52%) in the gastric phase occurs when the mass ratio of dietary fiber to tebuconazole-containing grapes is 0.1%.
Line 320 Remove „addition“
Answer: “addition” has been removed.
Lines 324-325 The sentence is incorrect. Cellulose is cellulose and pectin is pectin (it can be maybe classified as haemicellulose). There is no pectin in cellulose. Please change it.
Answer: Thanks for your clear observation, we have revised the sentence as “The soluble pectin in dietary fiber encapsulates the matrix, resulting in lower bioacces-sibility”.
Lines 338-339 The sentence: „The bioaccessibility of vegetable oil following dietary ingredient addition is complex (Fig. 2c)“ should be changed. I dont understand wht do you mean. You didnt measure the bioaccessibility of oil? Also, how can bioacessibility be complex? Rephrase.
Answer: Thanks for your comments, we have changed the sentence as” The bioaccessibility of tebuconazole after the vegetable oil addition was remarkable difference”.
Lines 340-341 The sentence: „In the intestinal phase, the addition amounts were 0.02, 0.05, 0.2, 0.4, and 1%, with significant differences“ should be changed. „addition amounts“ should be changed to „added amounts“. Also, what do you mean by „...with significant differences“? That they influenced bioaccessibility of tebuconazole with significant difference? Rephrase the sentence.
Answer: Thanks for your comments, we have changed the sentence as “The bioaccessibility of tebuconazole after the vegetable oil addition was remarkable difference (Fig. 2c). In the gastric phase, with increasing vegetable oil addition, bioaccessibility decreased from 57.78 to 46.67%. In the intestinal phase, the added amounts were 0.02, 0.05, 0.2, 0.4, and 1%, the bioaccessibility of tebuconazole with significant differences” in the manuscript.
Lines 343-344 Please explain what did you try to say with the sentence: „...tebuconazole has higher lipophilicity and separates water and oil 343 from the digestive juice“? It makes no sence. Rephrase or remove from the text.
Answer: Thanks for your comments, we have removed the sentence in the manuscript.
Line 344 What lipid membrane? Rephrase.
Answer: Thanks for your comments, we have rephrase the sentence as “Which may be that the oil-based surface activation of vegetable oils can effectively prevent pesticide migration”.
Lines 348-349 The sentence „Adding dietary fiber can significantly reduce the contribution of tebuconazole to bioaccessibility“ should be rephrased. How can tebuconazole contribute to its own bioaccessibility?
Answer: Thanks for your comments, we have revised the sentence as “Adding dietary fiber can significantly reduce the bioaccessibility of tebuconazole”.
Lines 358-359 The sentence: „To further investigate the process of tebuconazole absorption and transport in the 358 human body, a Caco-2 cell model was used“ must be changed. Caco-2 model can be used to assess intestinal transport/absorption but not the transport in the human body.
Answer: Thanks for your comments, we have revised the sentence as “To further investigate the process of tebuconazole absorption and transport in the intestinal tract, a Caco-2 cell model was used”
Lines 360-365 The paragraph can be removed. This is basic methodological issue and can optionally be moved to Materials and Methods section. Since it is a common knowledge, it is not necessary to keep it at all. In any case it should be removed from results. Only the last sentence, stating the TEER required for conducting the experiments can be left in Results.
Answer: Thanks for your comments, we have removed the basic methodological issue to the “Materials and Methods, 2.7.3. Transport studies”.
Line 366 The sentence „When tebuconazole is tested for cell viability...“ must be changed. One can not test tebuconazole for cell viability!
Answer: Thanks for your comments, we have revised the sentence as “Exogenous substances affect cell viability at a certain concentration. When testing for the effects of tebuconazole on cell viability, the same amount of solvent was used at the maximum dose”.
Lines 366-374 The whole paragraph should be significantly shorter and simplified. In short, you proved that acetonitrille or used tebuconazole concentrations used in the experiment didnt affect Caco-2 cell viability. One sentence like this is enough.
Answer: Thanks for your comments, we have shorter and simplified the paragraph as “Exogenous substances affect cell viability at a certain concentration. When testing for the effects of tebuconazole on cell viability, the same amount of solvent was used at the maximum dose. As show in Figure 3, different concentrations of acetoni-trile and tebuconazole used in the experiment didnt affect Caco-2 cell viability. The concentration of tebuconazole in the transfer test was higher than that of other triazole substances in the transfer test [40], which is likely due to the nature of tebuconazole [41]”.
Line 379 What is permeability directivity? You never mentioned it before nor you explained how it was calculated. In order to use it you need to explain the term and methodology it in materials and methods section
Thanks for your comments, we have added the formula of permeability directivity (PDR) calculation in the “2.8. Data analysis” as :
Permeability directivity (PDR)= Papp (AP-BL)/Papp (BL-AP)
Where the Papp (AP - BL) for the AP side switch to the side of the BL apparent permeability coefficient (cm/s), Papp (BL-AP) is the apparent permeability coefficient (cm/s) of BL side to AP side.
Line 382 Explain the meaning of AP-BL
Answer: AP means the apical (AP) side and the BL means the basolateral (BL) side, AP-BL means the transmembrane transport from AP side to the BL side. Which has been explaind in the manuscript.
Line 388 Remove „method“. Use „transport mechanism“ instead
Line 388-390 Remove the paragraph.
Answer: We have removed the paragraph “This method can determine the amount of tebuconazole fungicide delivered to the human body through Caco-2 cell parenchyma at the cellular level, providing a more multidimensional basis for the biological risk assessment of tebuconazole”.
Line 391. Change the title of Table 2. Doesnt make sense
Answer: Thanks for your comments, we have changed the title as “Changes of Tebuconazole Transport in Caco-2 Cell Model (n=3)”.
Please, also check several additional comments in the pdf version of the manuscript that i didnt mention here.
Answer: All the additional comments in the pdf version of the manuscript were also revised.

Reviewer 3 Report
The paper entitled ‘Bioaccessibility and intestinal transport of tebuconazole using table grape in vitro digestion models’ present interesting study results based on in vitro digestion models. Considering the fact that activity of active compounds depends on digestion process and changes in or organism, the in vitro digestion studies are important part of the part of analysis. On the whole, the paper is well prepared. The studies are interesting and novelty. Authors decided to use various methods which allowed to reliable assess obtained results. Authors used up-to-date references . Statistical analysis is provided. The paper enriches the science based on natural compounds.
1. Introduction section is too long. Please reduce it.
2. Study results are well presented but discussion should be enriched with more up-to-date data.
3. Conclusions should be also enriched with more information.
Author Response
The paper entitled ‘Bioaccessibility and intestinal transport of tebuconazole using table grape in vitro digestion models’ present interesting study results based on in vitro digestion models. Considering the fact that activity of active compounds depends on digestion process and changes in or organism, the in vitro digestion studies are important part of the part of analysis. On the whole, the paper is well prepared. The studies are interesting and novelty. Authors decided to use various methods which allowed to reliable assess obtained results. Authors used up-to-date references. Statistical analysis is provided. The paper enriches the science based on natural compounds.
- Introduction section is too long. Please reduce it.
Answer: Thanks for your comments, we have revised the introduction according to the comments.
- Study results are well presented but discussion should be enriched with more up-to-date data.
Answer: Thanks for your comments, we have revised the discussion section and updated some references according to the comments.
- Conclusions should be also enriched with more information.
Answer: Thanks for your suggestion, we have enriched the conclusions as follows:” This study compared tebuconazole bioaccessibility in several in vitro simulation methods and explicit the effects of different dietary components on the bioaccessibility of tebuconazole in grape. The SBRC method exhibited the highest bioaccessibility in the gastric and intestinal stages. When the pH, solid-liquid ratio, and time change, bioac-cessibility changes; for example, bioaccessibility gradually decreases with increasing pH. When time is used as a variable, the opposite is true. A solid-liquid ratio of 1:20 could provide higher bioaccessibility. The intake of dietary fiber could significantly reduce the absorption of tebuconazole, It is suggested that people should moderately strengthen the intake of dietary nutrients in their daily life. Furthermore, the Caco-2 cell model verified that tebuconazole is passively transported in the intestine. Our findings suggest that the bioaccessibility of pesticide residues needs to be taken into account when as-sessing the risk of dietary exposure to pesticides in the future.” in the manuscript.
